# Efficacy of cotrimoxazole (Sulfamethoxazole-Trimethoprim) as a salvage therapy for the treatment of bone and joint infections (BJIs)

**Laurene Deconinck[1], Aurélien Dinh[1], Christophe Nich[2], Thomas Tritz[3], Morgan Matt[1], Olivia Senard[1], Simon Bessis[1], Thomas Bauer[4], Martin Rottman[5], Jérome Salomon[1], Frédérique Bouchand[6], Benjamin Davido[1]***

**1** Service des Maladies Infectieuses, Centre Hospitalier Universitaire Raymond Poincaré, AP-HP, Garches, France, **2** Service d'Orthopédie, Centre Hospitalier Universitaire Raymond Poincaré, AP-HP, Garches, France, **3** Pharmacie Hospitalière, Centre Hospitalier Universitaire Ambroise Paré, AP-HP, Boulogne-Billancourt, France, **4** Service d'Orthopédie, Centre Hospitalier Universitaire Ambroise Paré, AP-HP, Boulogne-Billancourt, France, **5** Laboratoire de Microbiologie, Centre Hospitalier Universitaire Raymond Poincaré, AP-HP, Garches, France, **6** Pharmacie Hospitalière, Centre Hospitalier Universitaire Raymond Poincaré, AP-HP, Garches, France

\* benjamin.davido@aphp.fr

**Data Availability Statement:** All relevant data are within the manuscript.

**Funding:** I declare that authors did not received any grant or funding for this research.

## Abstract

### Introduction

Cotrimoxazole (Sulfamethoxazole-Trimethoprim, SXT) has interesting characteristics for the treatment of bone and joint infection (BJI): a broad spectrum of activity with adequate bone diffusion and oral and intravenous formulations. However, its efficacy and safety in BJIs are poorly documented and its use remains limited.

### Methods

We conducted a retrospective study in 2 reference centers for BJIs from 2013 to 2018 among patients treated with SXT for a BJI. Data were collected from patient's medical charts. Outcomes and adverse events were evaluated at day (D)7, D45 and D90.

### Results

We analyzed 51 patients with a mean age of 60 ± 20 (SD) years of which 76% presented with an orthopedic device infection (ODI). Gram-negative bacilli (GNB) were involved in 47% of BJIs (n = 24). Moreover, they were often polymicrobial infections (41%). Doses of SXT ranged from 800/160mg bid (61%; n = 31) to 800/160mg tid (39%; n = 20). Median SXT treatment duration was 45 days (IQR 40–45). SXT was part of a dual therapy in 84% of patients (n = 43), associated mainly with fluoroquinolones (n = 17) or rifampicin (n = 14). Outcome was favorable at D7 in 98% (n = 50), at D45 in 88.2% (n = 45) and at D90 in 78.4% (n = 40). The second agent combined with SXT was not an independent factor of favorable outcome (p = 0.97). Adverse events were reported in 8% (n = 4) of patients, with a median of 21 days (IQR 20–30) from SXT initiation and led to discontinuation (n = 3).

**Competing interests:** The authors have declared that no competing interests exist.

## Conclusion

SXT appears to be effective for treatment of BJIs as a salvage therapy, even in GNB or poly-microbial infection, including ODI. Further data are needed to confirm SXT efficacy as an alternative oral regimen in BJIs.

## Introduction

Bone and joint infections (BJIs) are a real concern in the context of an increase of orthopedic device implantations in an aging population with comorbidities. In addition to surgery, adequate antimicrobial therapy is required to efficiently treat BJIs [1]. Rifampicin is deemed to be the best antibiotic active against Gram positive cocci (GPC) in case of BJI, especially due to *Staphylococcus spp.*, in association with another antimicrobial agent [2,3]. One drawback with rifampicin is the frequent gastrointestinal side effects and the interaction with the p450 cytochromes, in particular in polymedicated elderly patients. Likewise, fluoroquinolones (FQ) are recommended in combination with rifampicin in staphylococcal BJI, or alone in infections due to Gram negative bacilli (GNB) [4–6]. This makes them particularly attractive due to a broad spectrum of activity with good bone diffusion and the possibility of oral administration [2]. Moreover, FQs have been shown to be as effective as intravenous beta-lactams for the treatment of osteomyelitis [7,8]. Thus, FQs took a central place in the treatment of BJIs with a high rate of success well-known since the 1990s [9]. However, in the last decades, resistance to FQs became increasingly prevalent [10]. In addition, a major drawback of prolonged FQ administration is the selection of Extended-Spectrum Beta-Lactamase (ESBL) producing *Enterobacteriaceae* [11] and their risk of induced-tendinitis but also neurotoxicity (around 2%) [12]. Another drawback of FQ therapy is the subsequent risk of *Clostridioides difficile* infection, in particular in the elderly. Therefore, alternative drugs to FQs are needed.

Cotrimoxazole (Sulfamethoxazole-Trimethoprim, SXT), is an inhibitor of folinic acid synthesis and has bacteriostatic activity against susceptible bacteria. SXT has some interesting characteristics for the treatment of BJIs. First, SXT is effective against both gram positive and gram negative bacteria, including MRSA [13–16]. Second, its bone diffusion is deemed to be adequate when high posologies are used, including when orally administered [17,18]. The efficacy of SXT in BJIs was first reported in the early 1970s [19]. Recent studies reported its efficacy especially in association with rifampicin in staphylococcal BJI [20,21], leading to the suggestion of SXT as an alternative to FQs in staphylococcal BJIs in the American, British and French guidelines [22–24]. Despite these encouraging data, SXT use remains limited, partly because of the related risk of adverse events, in particular cutaneous rash and haematotoxicity but also renal and hepatic impairment [25]. Therefore, new data to evaluate the efficacy and safety of SXT in BJIs are necessary, especially when other recommended oral agents cannot be prescribed.

The main objective of the study was to evaluate the effectiveness of SXT in BJIs after day 90. The secondary objective was to assess SXT-related adverse events.

## Methods

### Setting

A retrospective study was conducted in 2 reference centers for BJIs treatment located in the Greater Paris area, France (Ambroise Paré Hospital, Boulogne-Billancourt, and Raymond

Poincaré Hospital, Garches). Those 2 teaching hospitals share common staff with weekly pluridisciplinary meetings with an infectious disease specialist, an orthopedic surgeon, a microbiologist and a pharmacist. In our local guidelines, patients undergoing surgery for a BJI were administered empiric broad spectrum intravenous antimicrobial therapy post-operatively combining daptomycin and piperacillin-tazobactam in case of orthopedic device infection (ODI), or vancomycin and piperacillin-tazobactam in case of a native BJI, as appropriate. All adult patients admitted for a BJI in orthopedics and treated with SXT from January 2013 to April 2018 were enrolled in the study. Those centers managed 1568 BJIs during this period. Exclusion criteria were: age under 18 years, being infected by a microorganism non-susceptible to SXT, duration of SXT prescription of less than 10 days prescribed for the drainage of an abscess and a suppressive antimicrobial therapy (more than a year of therapy) or declining to participate in the study.

## Definitions

- Criteria for the diagnosis of acute bone and joint infection were based on clinical signs and symptoms of bone and joint infection associated with a positive bacteriological examination of blood, bone or joint fluid samples.

- Success was defined as the absence of local or systemic signs of infection, including delayed wound healing recorded in the medical chart, and a statistical diminution of the CRP value between admission and last follow-up consultation associated with absence of relapse. Cases who did not meet above-mentioned criteria were classified as "failure".

- Salvage therapy was defined by the inability to use a recommended regimen for the treatment of BJIs, notably a combination therapy with fluoroquinolones and rifampin for staphylococcal infection, due to a resistance mechanism or intolerance. In case of polymicrobial infection or potentially resistant organism (such as ESBL or cephalosporinase producing organisms), SXT was prescribed in combination therapy mainly to prevent the emergence of fluoroquinolones resistant mutants or broaden the antibiotic spectrum.

- Multidrug-resistant organisms included ESBL-producing *Enterobacteriaceae* and/or methicillin-resistant *Staphylococcus aureus* strains.

## Data collection

The following data were collected from patient's medical charts:

- Patient characteristics: age, sex, diabetes, smoking habits, peripheral vascular disease, allergy to antibiotic, Charlson score at admission, number of previous surgeries,

- Infection characteristics: orthopedic device (OD) (prosthetic joint, osteosynthesis, vertebral OD), site of infection, mono or polymicrobial infection, microorganism and mechanism of resistance, concomitant bacteremia, C-reactive protein (CRP) at diagnosis,

- Treatment characteristics: surgery (device retention, removal or replacement), empiric antimicrobial therapy, duration of intravenous antibiotic therapy, SXT use, route of administration, dosing and duration of treatment, mono or combination therapy with associated drugs.

- Outcomes were evaluated at day (D) 7 after the surgery, D45 and D90. The occurrence of death, adverse event, length of stay (LOS) and CRP at the last follow-up was collected. Later events were documented by telephone interviews.

## Statistical analysis

Descriptive results were expressed using median and interquartile range (IQR) for the continuous variables when appropriate, and in number with percentage for the categorical variables. Analyses were performed using Excel 2010 (Microsoft Corporation, Redmond, WA). Student's *t*-test was performed to analyze continuous data using GraphPad Prism v.7.0 (GraphPad Software Inc., La Jolla, CA). Statistical significance was defined as $p < 0.05$.

## Compliance with ethical standards and approval

All procedures performed in studies involving human participants were in accordance with the ethical standards of the institutional and/or national research committee and with the 1964 Helsinki Declaration and its later amendments or comparable ethical standards.

The local Ethics Committee was contacted and there was a waiver of any need for consent, linked to the retrospective nature of this study, since data have already been collected and thereafter data analyzed anonymously.

## Results

### Study population and infection characteristics

Overall, 124 patients were screened and are presented in the flow-chart (Fig 1). Fifty-one patients were included in the study which represents 3.2% of all the admission for BJI during this period. The median LOS was 10 days (IQR 8–17). Population and infection characteristics

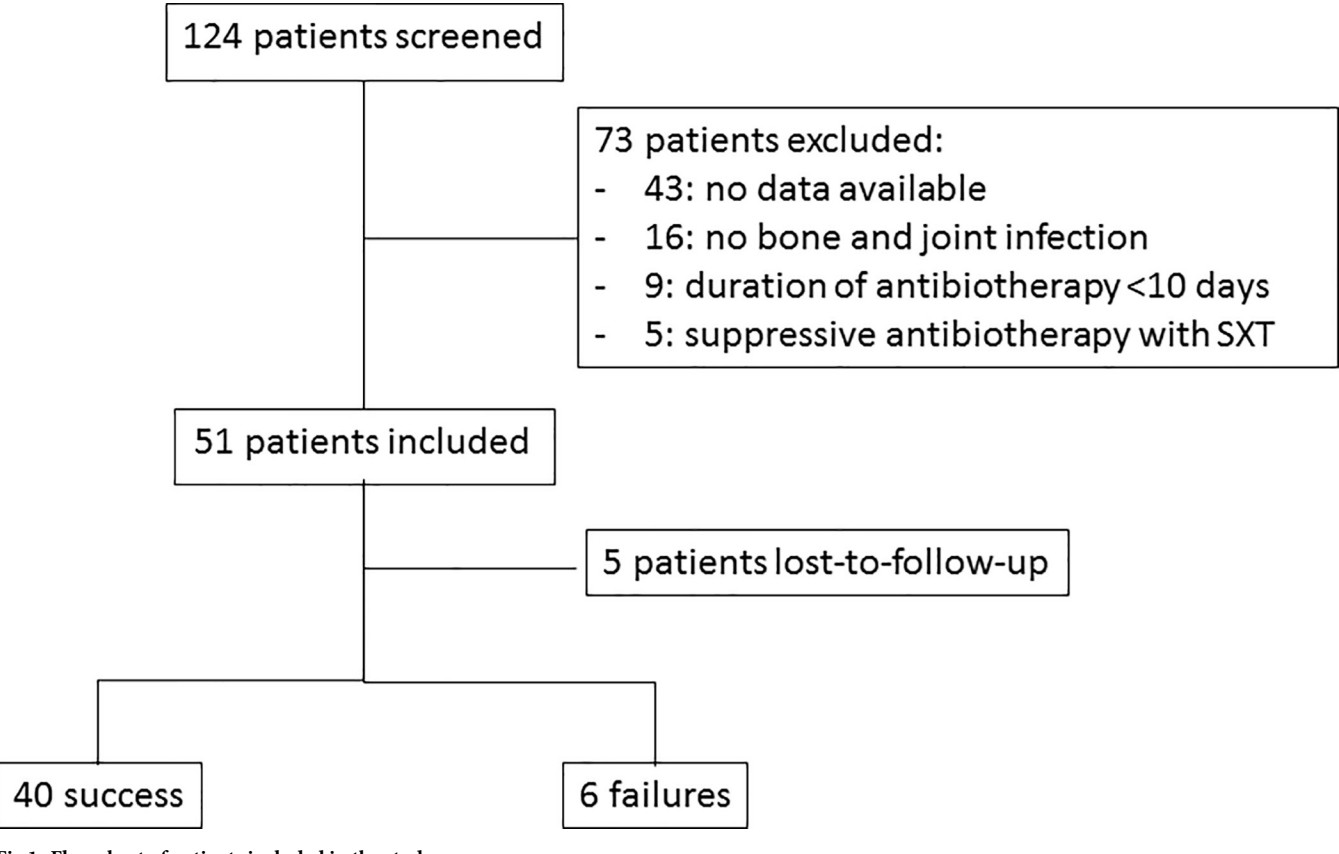

**Fig 1. Flow chart of patients included in the study.**

are detailed in Table 1. The sex ratio was 1 and the mean (± standard deviation (SD)) age was 60 ± 20 years. Twenty-one (41%) infections were polymicrobial, involving 2 to 4 bacterial species. BJIs were caused by Gram positive (GP) bacteria only (53%; n = 27), Gram negative (GN) bacteria (18%; n = 9) and 29% both GP and GN bacteria (n = 15). GPC included 39% coagulase-negative *Staphylococci* (CoNS) (n = 20), 27% methicillin-susceptible *Staphylococcus aureus* (MSSA) (n = 14) and 12% methicillin-resistant *S. aureus* (MRSA) (n = 6). GNB included 43% *Enterobacteriaceae* (n = 22) and 10% non-fermenter GNB (n = 5). *Enterobacteriaceae* were cephalosporinases producing species in 27% (n = 14) of cases and one case of ESBL-producing *E. coli*. Microorganisms are detailed in Table 2.

**Table 1. Population and infection characteristics.**

| | Total (N = 51) | | Success (N = 40) | | Failure (N = 11) | | P |
|---|---|---|---|---|---|---|---|
| **Population characteristics** | | | | | | | |
| Male, n(%) | 25 | (49) | 20 | (50) | 5 | (45) | NS |
| Age (years), median (IQR) | 60 | (44–77) | 60 | (44–76) | 70 | (42–76) | NS |
| Comorbidities, n(%) | 29 | (57) | 24 | (60) | 5 | (45) | NS |
| Diabetes, n(%) | 11 | (22) | 9 | (23) | 2 | (18) | NS |
| Smoking habits, n(%) | 21 | (41) | 18 | (45) | 3 | (27) | NS |
| Peripheral artery disease, n(%) | 8 | (16) | 6 | (15) | 2 | (18) | NS |
| Charlson score, median (min-max) | 3 | (0–8) | 3 | (0–8) | 4 | (0–8) | NS |
| Previous surgery, median (min-max) | 1 | (0–4) | 1 | (0–4) | 1 | (0–4) | NS |
| Allergy, n(%) | 5 | (10) | 3 | (8) | 2 | (18) | NS |
| *Beta-lactam, n(%)* | 3 | (6) | | | | | |
| *Fluoroquinolone, n(%)* | 2 | (4) | | | | | |
| **Infection characteristics** | | | | | | | |
| CRP at diagnosis (mg/l), median (IQR) | 77 | (26–111) | 67 | (21–108) | 78 | (59–133) | NS |
| Chronic infection, n(%) | 40 | (78) | 31 | (78) | 9 | (82) | NS |
| Type of infection | | | | | | | |
| *ODI, n(%)* | 39 | (76) | 29 | (73) | 10 | (91) | NS |
| *PJI, n(%)* | 28 | (55) | 20 | (50) | 8 | (73) | NS |
| *Osteosynthesis infection, n(%)* | 10 | (20) | 9 | (23) | 1 | (9) | NS |
| *Vertebral ODI, n(%)* | 1 | (2) | 0 | (0) | 1 | (9) | NS |
| *Osteomyelitis, n(%)* | 6 | (12) | 5 | (13) | 1 | (9) | NS |
| *Arthritis, n(%)* | 5 | (10) | 5 | (13) | 0 | (0) | NS |
| *Spondylodiscitis, n(%)* | 1 | (2) | 1 | (3) | 0 | (0) | NS |
| Site of infection | | | | | | | |
| *Lower limb, n(%)* | 42 | (82) | 32 | (80) | 10 | (91) | NS |
| *Knee, n(%)* | 13 | (25) | | | | | |
| *Hip, n(%)* | 12 | (24) | | | | | |
| *Leg, n(%)* | 8 | (16) | | | | | |
| *Ankle/foot, n(%)* | 7 | (14) | | | | | |
| *Femur, n(%)* | 2 | (4) | | | | | |
| *Upper limb, n(%)* | 7 | (14) | 7 | (18) | 0 | (0) | NS |
| *Elbow, n(%)* | 5 | (10) | | | | | |
| *Shoulder, n(%)* | 2 | (4) | | | | | |
| *Vertebral, n(%)* | 2 | (4) | 1 | (3) | 1 | (9) | NS |

CRP, C-reactive protein; ODI, orthopedic device infection; PJI, prosthetic joint infection.

**Table 2. Microbiological characteristics.**

| | Total (N = 51) | | Success (N = 40) | | Failure (N = 11) | | P |
|---|---|---|---|---|---|---|---|
| Polymicrobial, n(%) | 21 | (41) | 16 | (40) | 5 | (45) | NS |
| GP and GN bacterial infection, n(%) | 15 | (29) | 12 | (30) | 3 | (27) | NS |
| CGP, n(%) | 41 | (80) | 33 | (83) | 8 | (73) | NS |
| *CoNS, n(%)* | 20 | (39) | | | | | |
| *MSSA, n(%)* | 14 | (27) | | | | | |
| *MRSA, n(%)* | 6 | (12) | | | | | |
| *Streptococcus, n(%)* | 4 | (8) | | | | | |
| *Enterococcus, n(%)* | 4 | (8) | | | | | |
| GNB, n(%) | 24 | (47) | 18 | (45) | 6 | (55) | NS |
| *Enterobacteriaceae, n(%)* | 22 | (43) | | | | | |
| *Enterobacter sp, n(%)* | 10 | (20) | | | | | |
| *Escherichia coli, n(%)* | 5 | (10) | | | | | |
| *Klebsiella sp, n(%)* | 3 | (6) | | | | | |
| *Morganella morganii, n(%)* | 3 | (6) | | | | | |
| *Serratia sp, n(%)* | 3 | (6) | | | | | |
| *Proteus mirabilis, n(%)* | 2 | (4) | | | | | |
| *Group 3 Enterobacteriaceae, n(%)* | 14 | (27) | 9 | (23) | 5 | (45) | NS |
| *Non fermental BGN, n(%)* | 5 | (10) | 4 | (10) | 1 | (9) | NS |
| *Pseudomonas aeruginosa, n(%)* | 2 | (4) | | | | | |
| *Stenotrophomonas maltophilia, n(%)* | 2 | (4) | | | | | |
| *Acinetobacter sp, n(%)* | 1 | (2) | | | | | |
| Other, n(%) | 4 | (8) | 3 | (8) | 1 | (9) | NS |
| *Propionibacterium acnes, n(%)* | 2 | (4) | | | | | |
| *Corynebacterium sp, n(%)* | 2 | (4) | | | | | |
| MDR bacteria, n(%) | 7 | (14) | 5 | (13) | 2 | (18) | NS |
| *MRSA, n(%)* | 6 | (12) | | | | | |
| *ESBL, n(%)* | 1 | (2) | | | | | |
| Bacteraemia, n(%) | 1 | (2) | 1 | (3) | 0 | (0) | NS |

GP, gram positive; GN, gram negative; CGP, cocci gram positive; GNB, gram negative bacilli; CoNS, coagulase-negative *Staphylococci*; MSSA, methicillin-susceptible *Staphylococcus aureus*; MRSA, methicillin-resistant *Staphylococcus aureus*; MDR, multi-drug resistant; ESBL, extended-spectrum beta lactamase. Of note, the total number of microorganisms exceeds 100% as there are polymicrobial infections.

Overall, 25 patients had organisms which were resistant to fluoroquinolones and 5 patients were considered intolerant to the oral agents, notably among elderly patients related to confusional states.

## Surgical management

All patients underwent surgery during the course of therapy. The procedure consisted of debridement on native joint in 12 cases (24%) debridement and implant retention in 11 cases (22%) and debridement with OD removal in 28 cases (55%) (Table 3). Eight ODIs presented acutely versus 31 chronic ODIs. Debridement and implant retention was performed in 26% (n = 8) of cases vs debridement with OD removal in 74% (n = 23).

## Antimicrobial regimens

Median duration of first-line IV treatment was 7 days (ranging from 5 to 10 days).

**Table 3. Surgical management and antimicrobial regimen.**

| | Total (N = 51) | | Success (N = 40) | | Failure (N = 11) | | P |
|---|---|---|---|---|---|---|---|
| **Surgical management** | | | | | | | |
| Surgery, n(%) | 51 | (100) | 40 | (100) | 11 | (100) | NS |
| *Debridement on native joint, n(%)* | 12 | (24) | 11 | (28) | 1 | (9) | NS |
| *Debridement and implant retention, n(%)* | 11 | (22) | 9 | (23) | 2 | (18) | NS |
| *Debridement with OD removal, n(%)* | 28 | (55) | 20 | (50) | 8 | (73) | NS |
| *1 stage change, n(%)* | 19 | (68) | 14 | (35) | 5 | (45) | NS |
| *2 stages change, n(%)* | 1 | (4) | 1 | (3) | 0 | (0) | NS |
| *No implantation, n(%)* | 8 | (29) | 5 | (13) | 3 | (27) | NS |
| **Medical management** | | | | | | | |
| Empiric antiimicrobial therapy, n(%) | 50 | (98) | 39 | (98) | 11 | (100) | NS |
| *Beta-lactam, n(%)* | 49 | (96) | 38 | (95) | 11 | (100) | NS |
| *Piperacillin-tazobactam, n(%)* | 43 | (84) | | | | | |
| *Other, n(%)* | 6 | (12) | | | | | |
| *Anti-MRSA ATB, n(%)* | 46 | (90) | 36 | (90) | 10 | (91) | NS |
| *Vancomycin, n(%)* | 23 | (45) | | | | | |
| *Daptomycin, n(%)* | 23 | (45) | | | | | |
| *Other, n(%)* | 4 | (8) | 3 | (8) | 1 | (9) | NS |
| *Intra-venous ATB duration (days), median (IQR)* | 6 | (5–7) | 6 | (5–7) | 6 | (5–8) | NS |
| SXT prescription modalities | | | | | | | |
| *Daily dose, n(%)* | | | | | | | |
| *800/160mg bid, n(%)* | 31 | (61) | 25 | (63) | 6 | (55) | NS |
| *800/160mg tid, n(%)* | 20 | (39) | 15 | (38) | 5 | (45) | NS |
| *Oral intake, n(%)* | 51 | (100) | 40 | (100) | 11 | (100) | NS |
| *Number of associated ATB, n(%)* | | | | | | | |
| *1, n(%)* | 43 | (84) | 36 | (90) | 7 | (64) | NS |
| *≥2, n(%)* | 8 | (16) | 4 | (10) | 4 | (36) | NS |
| *Associated ATB in dual therapy, n(%)* | | | | | | | |
| *FQ, n(%)* | 17 | (40) | 14 | (35) | 3 | (27) | NS |
| *RMP, n(%)* | 14 | (33) | 12 | (30) | 2 | (18) | NS |
| *Other, n(%)* | 12 | (28) | 10 | (25) | 2 | (18) | NS |
| *Oral ATB duration (days), median (IQR)* | 45 | (40–45) | 45 | (40–45) | 37 | (34–44) | NS |
| Total ATB duration (days), median (IQR) | 47 | (45–51) | 48 | (45–51) | 42 | (41–50) | NS |

OD, orthopedic device; ATB, antibiotic; MRSA, methicillin-resistant *Staphylococcus aureus*; SXT, Sulfamethoxazole-trimethoprim; RMP, rifampicin; FQ, fluoroquinolone.

All patients received an oral regimen with SXT as part of a combination therapy, with 43 (84%) dual therapy and 8 (16%) tritherapy or more (detailed in Table 3). Of the eight patients treated with three or more antimicrobial agents, 88% were polymicrobial (n = 7) involving both GPC and GNB in 38% (n = 3).

The associated drugs used in dual therapy were mainly FQ or rifampicin (72%, n = 31). Among patients treated with a dual therapy including a FQ (n = 17), 11 (65%) were due to a cephalosporinase producing *Enterobacteriaceae*. Conversely, 12 (28%) cases received neither rifampicin nor FQ in bitherapy with SXT. The regimen included beta-lactams (n = 4) (amoxicillin-clavulanic acid (n = 3) and cefepime (n = 1)), clindamycin (n = 4), daptomycin (n = 2), vancomycin (n = 1) or linezolid (n = 1).

## Outcomes

The median duration of follow-up was 126 days (IQR 99–185); 5 (9.8%) patients were lost to follow-up at D90, including 2 cases before D45.

The median CRP level at the last follow-up was 10 mg/L (IQR 2–24) and lower than at the admission (see Table 1) (p<0.0001). In our center, the CRP value was considered normal at a value of ≤5 mg/L and was found in 21 (41.2%) patients at the last follow-up consultation.

Considering those who were lost to follow-up as failures, outcomes were favorable at D7 in 98% (n = 50), at D45 in 88.2% (n = 45) and at D90 in 78.4% (n = 40). If we exclude those who were lost to follow-up from the final analyses (best-case scenario), outcomes were favorable at D7 in 98% (n = 50/51), at D45 in 91.8% (n = 45/49) and at D90 in 87.0% (n = 40/46). Outcomes at D90 depending on the drug used in combination therapy with SXT were comparable between groups (p = 0.97) and are detailed in Fig 2.

In univariate analyses, infection with a cephalosporin producing *Enterobacteriaceae* (n = 14) was not significantly associated with a worse outcome at D90 with a 64.3% (n = 9) cure rate vs other groups 83.8% (n = 31) (p = 0.15). Of note, 78.6% (n = 11) were treated with a dual therapy containing SXT/FQ.

No patient died nor was admitted to intensive care unit in the course of treatment. Twenty-one patients (41%) had a limitation in their function at the end of follow-up.

## Adverse events

Adverse events were reported in 4 (8%) patients, with a median time of 21 days (IQR 20–30) from SXT introduction. Adverse events consisted of a mild hepatitis (values up to 3 times above the normal), a short duration watery diarrhea, a cutaneous maculopapular skin rash and an isolated fever in 1 case each. No renal failure was diagnosed during follow-up. SXT was interrupted in 6% (n = 3): 1 patient had replacement with a switch to another antimicrobial therapy (n = 1) and 2 antibiotic discontinuations were noted. Three of them were lost to follow-up at D90.

## Discussion

This retrospective study documents numerous patients treated with a combination therapy based on SXT for the treatment of a polymicrobial BJI (41%). We observed a favorable outcome at D90 in 78.4%; nevertheless adverse events led to SXT discontinuation in 6% of patients. It is noteworthy that when used in dual therapy, the second agent combined with SXT did not impact the overall success rate.

A success rate of less than 80% can appear disappointing but it rises up to 87% in the best case scenario. This rate is gratifying if we consider the higher proportion than usual of these particular prosthetic joint infections due to polymicrobial and Gram-negative bacteria [10,26]. Moreover it concerns elderly and fragile patients with frequent comorbidities who underwent previous surgeries which failed initial therapy (as illustrated in Table 1) and therefore required the expertise of a reference center.

This rate of favorable outcome is slightly higher than previously reported in the literature in GPC and GNB BJIs treated with SXT. Indeed, Fica *et al*. reported a success rate of 61% (n = 23) with 9 failures requiring surgery [27]. In another study regarding early ODI due to GCP and GNB, Torenero *et al*. support the fact that SXT is inappropriate due to a high rate of failure but in a small sample size of 7 patients (including 5 due to GPC and 2 to GNB) [28].

Our results are similar to those more recently reported in GPC BJIs. Euba *et al*. showed that SXT in combination with rifampicin was as effective as intravenous cloxacillin in the treatment of 28 chronic staphylococcal osteomyelitis, with 88.9% success in intention to treat and 91.7%

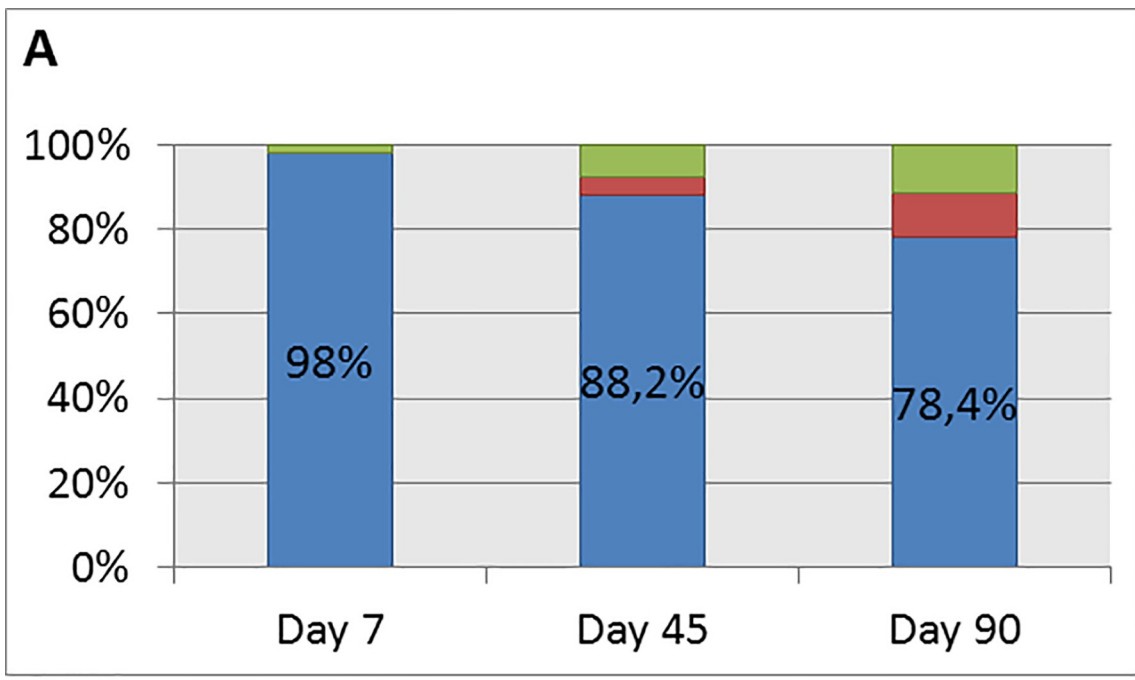

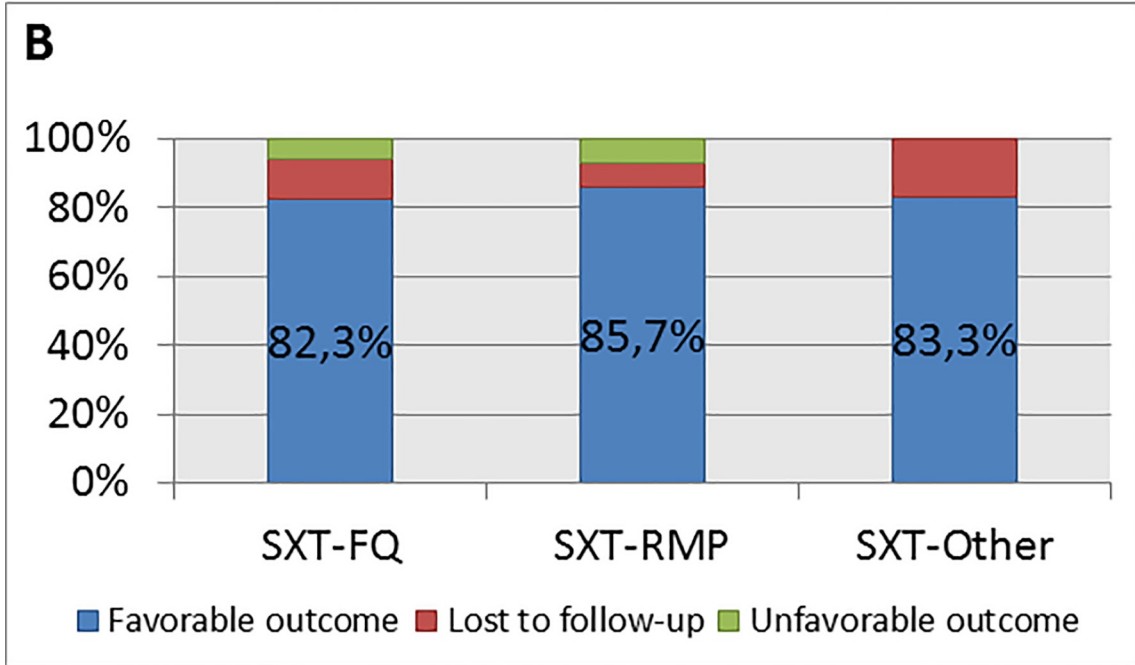

**Fig 2. Outcomes at day 7, day 45 and day 90, overall (2A) and at day 90 depending on the drug associated with SXT in dual therapy (2B).**

in per protocol analyses [29]. Moreover, Nguyen *et al.* reported a success rate of 78.6% at the end of the treatment and 76.9% at 2 years. This study was conducted among 26 patients with a BJI treated with SXT in association with rifampicin. This latter study showed similar outcomes using this regimen based on SXT versus a combination of rifampicin plus linezolid [30]. Likewise, Harbarth *et al.* described a 85.7% success rate (n = 75) using SXT in association with rifampicin for the treatment of MRSA BJI in comparison with 55.6% when using linezolid

alone [31]. Finally, in pediatrics, Messina *et al*. reported a success rate of 100% (n = 20) using a monotherapy of SXT for the treatment of acute *Staphylococcus aureus* osteomyelitis [32].

Of note, one major point pleading for the use of SXT over FQ is that MRSA strains seem to remain susceptible to SXT even after a SXT exposure [33]. However, physicians should be wary that there is a possible risk of emergence of resistance against rifampicin when prescribed in dual therapy with SXT, despite a combination therapy [34]. Nevertheless, it should be noted that those concerns are based on *in vitro* data and must be discussed in the light of clinical findings.

In addition, data are scarce concerning GNB prosthetic joint infections, which represents 15% of BJIs and are known to be difficult to treat [35]. Nevertheless, our data did not show any statistical differences in terms of outcomes between cephalosporinase producing *Enterobacteriaceae* and the other cases, likely because of an underpowered dataset (n = 51). A combination therapy of SXT with a FQ in these patients was associated with a favorable outcome in 82.3%. Although the systemic use of a combination therapy against GNB including cephalosporinase producers might have been overcautious, it may have helped to preserve FQs acquiring resistance.

In addition, median duration of treatment was 6 weeks which is generally compliant according to the French guidelines [22] and promoting the appropriate use of antibiotic duration. In such prolonged therapy, an efficient oral regimen can be helpful to reduce LOS. Indeed, our reported LOS was quite short with a median of 10 days. Previously published studies have also reported a diminution of the LOS for BJI while using SXT [27,30]. Likewise, although costs have not been studied, some literature has already reported a diminution of costs when prescription included SXT [26,30,36].

Even if the reason to prescribe SXT over other molecules was not clearly stated in the medical records, most of the prescriptions were supported by the resistance to either FQ or rifampicin. Indeed, allergy was not the main reason for prescribing SXT since only a few patients presented allergy to FQ and none to rifampicin.

Overall, SXT's broad spectrum activity could be of interest in various populations, including elderly patients with polymicrobial infections, previously exposed to usual classes of antibiotics (i.e. FQ and rifampicin).

Interestingly, the doses of SXT we used were frequently lower than recommended by the French (3200/640mg) and US guidelines (8 mg/kg of trimethoprim) [22,23], but also lower than the ones reported in previous cohorts of BJIs treated with SXT [27,29–31]. This can be explained by the fact that physicians did not want to use high doses considering the elderly population with an increased risk of side effects. Although it can be argued that lower doses can be responsible for antibiotic selective pressure, our prescriptions were performed as combination therapy, thereby reducing the potential risk of the appearance of mutants.

SXT is responsible for various adverse events such as cutaneous rash, gastrointestinal disorders, hepatitis, cytopenia [25] or acute renal failure [37,38]. In our study, adverse events were reported in only 8% of patients, leading to SXT interruption in 6%. A similar rate was reported in previous studies concerning staphylococcal BJI with 12% adverse events and 7% SXT discontinuation due to adverse events [29,31]. Likewise, Valour *et al*. reported a comparable rate of 15% of adverse events in patients treated with various other antibiotics for methicilllin-susceptible *Staphylococcus aureus* BJI [39]. However, Nguyen *et al*. reported a higher rate of 46% of adverse events using SXT [30]. These results can partly be explained by the long mean duration of therapy of 17.8 weeks. Likewise, Fica *et al*. reported 43% of SXT discontinuation mainly due to hyperkalemia in patients treated with drugs active against the renin-angiotensin system [27]. Those adverse events must be balanced with the new alert of aortic aneurysm and

dissection due to the prolonged used of fluoroquinolones (>14 days) [40], that is a serious threat to consider in such population of elderly and comorbid patients suffering from BJIs.

Finally, our study presents some limitations. First, it is a retrospective study with a relatively limited sample size; thereby its results might not be extrapolated to other healthcare facilities. Second, the infections included are heterogeneous with OD and native BJI caused by GPC and GNB infections. However, it reinforces the usefulness of SXT when clinicians are dealing with polymicrobial infections. Third, we faced a substantial number of patients lost to follow-up despite a relatively short median duration of follow-up for a BJI, considering that infection cure is usually defined after a one-year follow-up. However, individuals who were lost to fol-low-up were interpreted as failures to avoid an over-estimation of the success rate in final analyses and this is what we observed in real-life.

## Conclusion

Cotrimoxazole appears to be effective for the treatment of BJIs. Its broad spectrum activity makes it particularly interesting in polymicrobial infections with GPC and GNB as illustrated in our work. It can prove helpful as a salvage therapy when the preferred combination therapy with FQ or rifampicin is unusable, but also in case of deadlock situations requiring intravenous and prolonged therapy. Although the described rate of reported adverse events was acceptable, physicians should be wary that most patients who experienced adverse events required discontinuation of their therapy. We believe our work will encourage reference centers taking into consideration the use of SXT. Further prospective data are needed to confirm SXT efficacy as an alternative regimen in common BJIs.

## Acknowledgments

Authors would like to thank their colleagues, particularly Pr Anne-Claude Cremieux and Dr Pierre de Truchis for their unfailing support.

## Author Contributions

**Conceptualization:** Laurene Deconinck, Aurélien Dinh, Simon Bessis, Thomas Bauer, Jérome Salomon, Frédérique Bouchand, Benjamin Davido.

**Data curation:** Jérome Salomon.

**Formal analysis:** Frédérique Bouchand, Benjamin Davido.

**Investigation:** Morgan Matt, Olivia Senard, Thomas Bauer, Benjamin Davido.

**Methodology:** Thomas Tritz, Jérome Salomon, Frédérique Bouchand, Benjamin Davido.

**Project administration:** Aurélien Dinh, Benjamin Davido.

**Supervision:** Martin Rottman, Frédérique Bouchand, Benjamin Davido.

**Validation:** Frédérique Bouchand, Benjamin Davido.

**Visualization:** Frédérique Bouchand.

**Writing – original draft:** Laurene Deconinck, Benjamin Davido.

**Writing – review & editing:** Christophe Nich, Simon Bessis, Martin Rottman.

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
