## [Decision Letter · Decision Letter 0]

16 Aug 2019

PONE-D-19-17215

Efficacy of cotrimoxazole (Sulfamethoxazole-Trimethoprim) as a salvage therapy for the treatment of bone and joint infections (BJI)

PLOS ONE

Dear Dr. Davido,

Thank you for submitting your manuscript to PLOS ONE. After careful consideration, we feel that it has merit but does not fully meet PLOS ONE’s publication criteria as it currently stands. Therefore, we invite you to submit a revised version of the manuscript that addresses the points raised during the review process.

Please address all the peer reviewer comments . Some substantive issues have been raised with which I agree. and need to be addressed. In addition there are several Editorial comments that need to be addressed which are outlined below. 

It is not clear how patients were selected given that many patients may have been referred and some were treated with TMP-SMX . Please clarify what proportion of patients were treated with this agent  aby nd how the selection was made ie a convenience sample or some other method.

The definitions need more details and especially "diminution of CRP - was a normal value required ? Please be as objective as possible.

Please clarify within the manuscript the number of cases considered resistant or intolerant  to the usual recommended regimen of a FQ  plus rifampin -and then reconcile why in some cases TMP-SM|X was combined with a FQ.. 

There were a large number of patients excluded due to " no data" which deserves more explanation ie was it missing or not available ? 

Please indicate if the Ethics Committee was contacted and there was a waiver of any need for consent. 

The susceptibility of the organisms listed should be provided to both the FQs and TMP-SMX. 

Define multidrug resistant.

Add something to the legend in Table 2 about the microbes and that they do not add to 100% given the multiple isolates.

Please explain  why some cases had dual therapy with both a FQ and TMP-SMX. What was the rationale ?

Please tell us the normal values for the CRP and how many patients achieved a normal CRP.

As mentioned by the peer reviewer the use of intention to treat and per protocol is usually reserved for RCTs so please rephrase this or if some other terms are used explain and introduce them in the Methods.

Please define what represents hepatitis, diarrhea and cutaneous rash in more explicit terms so the readers can determine what is mean by the terms. .

There are a large number of typographical and grammatical errors which need correcting throughout the manuscript.

The references have many errors and there are case issues which are not consistent in the listing of the references and no Latin terms have italics eg names of microbes

We would appreciate receiving your revised manuscript by Sept 30, 2019. To enhance the reproducibility of your results, we recommend that if applicable you deposit your laboratory protocols in protocols.io, where a protocol can be assigned its own identifier (DOI) such that it can be cited independently in the future. For instructions see: http://journals.plos.org/plosone/s/submission-guidelines#loc-laboratory-protocols

We look forward to receiving your revised manuscript.

Kind regards,

John Conly, MD

Academic Editor

PLOS ONE

Journal Requirements:

1. Thank you for including your funding statement; "no"

Please provide an amended Funding Statement that declares *all* the funding or sources of support received during this specific study (whether external or internal to your organization) as detailed online in our guide for authors at http://journals.plos.org/plosone/s/submit-now.  

Please state what role the funders took in the study.  If any authors received a salary from any of your funders, please state which authors and which funder. If the funders had no role, please state: "The funders had no role in study design, data collection and analysis, decision to publish, or preparation of the manuscript."

2. Thank you for including your competing interests statement; "no"

Reviewers' comments:

Reviewer's Responses to Questions

**Comments to the Author**

1. Is the manuscript technically sound, and do the data support the conclusions?

Reviewer #1: No

2. Has the statistical analysis been performed appropriately and rigorously? 

Reviewer #1: I Don't Know

3. Have the authors made all data underlying the findings in their manuscript fully available?

Reviewer #1: No

4. Is the manuscript presented in an intelligible fashion and written in standard English?

Reviewer #1: No

5. Review Comments to the Author

Reviewer #1: This is an interesting observational paper but the methods and limited data don't allow confidence in the conclusions proposed. The majority of patients received dual drug therapy which makes conclusions about the efficacy of cotrimoxazole alone for these infections questionable. A 10% loss to follow-up could substantially affect success rates and I am concerned about an additional 9 patients excluded due to duration of therapy less than 10 days and 5 excluded due to suppressive therapy. The reasons for therapy less than 10 days are not provided - was this a reflection of treatment failure? How it was determined in a retrospective chart review that treatment was for suppression rather than curative therapy is also not made clear either. These 15 patients could represent additional treatment failures but this can't be determined from the information available.

Success criteria are listed as "the absence of local or systemic sign of infection, including

delayed wound healing, and the diminution of CRP and the absence of relapse". It isn't clear how in a retrospective chart review a determination was made about the "local or systemic signs of infection" or "delayed wound healing". What was the quality of documentation on these features in the charts reviewed? How was relapse defined? How subjective was this analysis? Were there specific definitions based on what was found in the charts for these criteria?

The small sample size combined with 10% loss to follow-up and the 15 excluded patients above combined with the retrospective nature of the review and possibly very subjective criteria for success when determined in a chart review make any conclusions suspect.

The conclusion states "SXT appears to be effective for treatment of BJI as a salvage therapy" and in description salvage therapy was referred to as "the impossibility to use recommended regimen for the treatment of BJI, notably a combination therapy with fluoroquinolones and rifampin, either due to a resistance mechanism or an intolerance."

The paper fails to provide adequate data to explain why cotrimoxazole might have been chosen in these patients and specifically states that "allergy was not the main reason for prescribing SXT since only a few patients presented allergy to FQ and none to rifampicin" and further suggests that the reason cotrimoxazole was chosen was not necessarily stated in the medical records. Normally salvage therapy implies a failed trial of usual therapy before a switch to the "salvage" therapy. This is not clarified in the paper. Neither the definition of salvage therapy nor the description of why cotrimoxazole was chosen are appropriate to reach the conclusion stated.

Use of terms "intention to treat" and "per protocol" in description of the analysis is inappropriate as it implies a prospective randomized experimental design rather than a retrospective review. Therre was no "protocol" when the patients were started on therapy.

6. PLOS authors have the option to publish the peer review history of their article (what does this mean?). If published, this will include your full peer review and any attached files.

Reviewer #1: Yes: Andrew L S Pattullo

---

## [Author Response · Author response to Decision Letter 0]

21 Aug 2019

First, we would like to thank the editor and the reviewer for their relevant suggestions and comments that have been considered and improved the overall revised manuscript. We hope you will find this version clarified and useful for the readers.Please find attached the rebuttal letter with the correction we made in the revised manuscript.

Regards,

---

## [Editor Report · Decision Letter 1]

11 Sep 2019

[EXSCINDED]

PONE-D-19-17215R1

Efficacy of cotrimoxazole (Sulfamethoxazole-Trimethoprim) as a salvage therapy for the treatment of bone and joint infections (BJIs)

PLOS ONE

Dear Dr. Davido,

Thank you for submitting your revised manuscript to PLOS ONE which has been substantially improved . However there remain a number of issues, mainly grammatical and syntax that need correcting. Therefore, we invite you to submit a revised version of the manuscript that addresses the points raised below.

Abstract

- Should be “ a broad spectrum of activity…”

- The sentences “ We noted 76% of  orthopedic device infection (ODI). Infections were mostly polymicrobial (41%), including 47 % of gram-negative bacilli (GNB). “ and “ The drug associated with SXT was not an independent factor of success (p=0.97).”  are grammatically poor and need correcting

Prefer wording of  “ alternative oral  regimen in BJIs.” In the last sentence

Introduction

Better wording would be as follows in the sentences containing the following phrases

 “ ….concern in the  context of an increase ….”

“ to efficiently treat BJIs “

“  p450 ctyochromes “

“ a  major drawback of prolonged FQ administration is the”  

Therefore, alternative drugs to FQs are needed. (remove the remainder of the sentence ie if we want to apply a case

“The  efficacy of SXT  in BJIs was first reported in the early  early 1970s

“ …………when other  recommended oral  agents cannot be prescribed.”

Please use italics for Latin terms eg organism names throughout the manuscript ie *Staphylococcus* species

Methods

Better wording would be as follows in the sentences containing the following phrases

“ ………..were administered empiric broad spectrum intravenous

antimicrobial therapy post-operatively combining ……”

All adults patients should be “ all adult patients “

Please spell out the abbreviated ODI in full for first time use in the body of the manuscript

Should be “……..or systemic signs of infection …” rather than sign

Correct to  “ ….wound healing recorded in the medical chart,……”

Should be “ diminution of the  CRP value…. “

Should be “ … defined by the inability to use a recommended ….”

Should be “ … or potentially resistant organism (such as ESBL or

 cephalosporinase producing  organisms…..   broaden the antibiotic spectrum.

Should be “ ESBL-producing Enterobacteriaceae and/or ….”

Should be “ empiric” rather than empirical

Should be “The local Ethics Committee was contacted ………”

Results

Should be “ patients were included in the study which represents …. “

Should be “…..(n=14) and 12% methicillin-resistant…”

Should be worded better as  “ Overall, 25 patients had organisms which were resistant to fluoroquinolones and 5 patients  were considered intolerant to the oral agents,  notably among elderly patients related to confusional states. “

Should be “……..surgery during  the course of therapy.”

Should be “ ……………. The procedures consisted of …….” 

Should be “ ………….was performed in 26%  (n=8) of cases vs debridement….”

Should be “…. Table 3. Surgical management and antimicrobial regimen.’ 

Same for the subtitle – better as Antimicrobial Regimens “

Please use another term other than “relay” for the use of an oral regimen

Should be “In our center, the CRP value was considered normal at a value of  ≤5

 mg/L and was found in 21 (41.2%) patients at the last follow-up consultation.”

Should be “………….not significantly associated……………..”

Should be “Adverse events consisted of  a mild hepatitis (values up to 3 times

above the normal), a short duration watery diarrhea,  a cutaneous maculopapular skin

 rash and an isolated fever in 1 case each. No renal failure was diagnosed during follow-up.

 SXT was interrupted in 6% (n=3): 1 patient had replacement with a switch to another antimicrobial

 therapy (n=1) and 2 antibiotic discontinuations were noted. “

Discussion

Better wording would be as follows in the sentences containing the following phrases

Should be “ This retrospective study documents numerous patients treated with a

combination therapy based on SXT for the treatment of a polymicrobial BJI”

Should be “ ………. can appear disappointing but it rises up to 87%  in the best case scenario.”

Should be “ This rate is gratifying  if we consider the higher proportion than usual of these particular prosthetic joint infections due to polymicrobial and Gram negative bacteria.”

Should be “  Moreover it concerns elderly and fragile patients with frequent comorbidities who underwent previous surgeries which failed initial therapy and therefore required the expertise of a reference center.”

Should be “ This latter study showed similar  outcomes using this regimen based on SXT versus………..”

Should be “ In addition, median duration of treatment was 6 weeks which is generally compliant according to the French guidelines [22] and promoting the appropriate use of antibiotic duration.”

Should be “ Likewise, although costs have not been studied, some  literature has already reported a diminution of costs when…….” And there should be a reference added to support .

The phrase “………but also than the ones reported in previous cohorts of BJI treated with SXT………… “ is missing something – please correct

Should be “ ………. For  antibiotic selective pressure,”

Eliminate the sentence “A particular attention to renal function and potassium level must be paid in these patients especially when using high dosing of SXT. “ It adds very little and helps to tighten the manuscript.

Conclusion

Eliminate “……….in common daily encountered practice………….”

References

There are a few items to correct with respect to case in the references eg the last 2 and other refs have the title of the paper all in capital letters – please correct

We would appreciate receiving your revised manuscript by %Sept 30, 2019. To enhance the reproducibility of your results, we recommend that if applicable you deposit your laboratory protocols in protocols.io, where a protocol can be assigned its own identifier (DOI) such that it can be cited independently in the future. For instructions see: http://journals.plos.org/plosone/s/submission-guidelines#loc-laboratory-protocols

We look forward to receiving your revised manuscript.

Kind regards,

John Conly, MD

Academic Editor

PLOS ONE

---

## [Author Response · Author response to Decision Letter 1]

23 Sep 2019

Dear Dr. Davido,

Thank you for submitting your revised manuscript to PLOS ONE which has been substantially improved . However there remain a number of issues, mainly grammatical and syntax that need correcting. Therefore, we invite you to submit a revised version of the manuscript that addresses the points raised below.

First, we would like to thank the editor (John Conly) for his finesse and suggestions of clarification. All the comments have been taken into consideration. We believe this version is now ready for publication.

Abstract

- Should be “ a broad spectrum of activity…”

- The sentences “ We noted 76% of orthopedic device infection (ODI). Infections were mostly polymicrobial (41%), including 47 % of gram-negative bacilli (GNB). “ and “ The drug associated with SXT was not an independent factor of success (p=0.97).” are grammatically poor and need correcting

We rephrased it as follows: “Gram-negative bacilli (GNB) were involved in 47% of BJIs (n=24). Moreover, they were often polymicrobial infections (41%)” And 

“The second agent combined with SXT was not an independent factor of favorable outcome (p=0.97).”

Prefer wording of “ alternative oral regimen in BJIs.” In the last sentence

 Corrected

Introduction

 Better wording would be as follows in the sentences containing the following phrases

 “ ….concern in the context of an increase ….”

“ to efficiently treat BJIs “

“ p450 ctyochromes “

“ a major drawback of prolonged FQ administration is the” 

Therefore, alternative drugs to FQs are needed. (remove the remainder of the sentence ie if we want to apply a case

“The efficacy of SXT in BJIs was first reported in the early early 1970s

“ …………when other recommended oral agents cannot be prescribed.”

 Please use italics for Latin terms eg organism names throughout the manuscript ie Staphylococcus species

 Corrected

Methods

 Better wording would be as follows in the sentences containing the following phrases

 “ ………..were administered empiric broad spectrum intravenous

antimicrobial therapy post-operatively combining ……”

 All adults patients should be “ all adult patients “

Please spell out the abbreviated ODI in full for first time use in the body of the manuscript

 Corrected

Should be “……..or systemic signs of infection …” rather than sign

Correct to “ ….wound healing recorded in the medical chart,……”

Should be “ diminution of the CRP value…. “

Should be “ … defined by the inability to use a recommended ….”

Corrected

Should be “ … or potentially resistant organism (such as ESBL or

 cephalosporinase producing organisms….. broaden the antibiotic spectrum.

Should be “ ESBL-producing Enterobacteriaceae and/or ….”

Should be “ empiric” rather than empirical

Should be “The local Ethics Committee was contacted ………”

Corrected

Results

 Should be “ patients were included in the study which represents …. “

 Should be “…..(n=14) and 12% methicillin-resistant…”

Should be worded better as “ Overall, 25 patients had organisms which were resistant to fluoroquinolones and 5 patients were considered intolerant to the oral agents, notably among elderly patients related to confusional states. “

Should be “……..surgery during the course of therapy.”

Should be “ ……………. The procedures consisted of …….” 

Should be “ ………….was performed in 26% (n=8) of cases vs debridement….”

Should be “…. Table 3. Surgical management and antimicrobial regimen.’ 

Same for the subtitle – better as Antimicrobial Regimens “

Corrected

Please use another term other than “relay” for the use of an oral regimen

We simply used the term oral regimen.

Should be “In our center, the CRP value was considered normal at a value of ≤5

 mg/L and was found in 21 (41.2%) patients at the last follow-up consultation.”

Should be “………….not significantly associated……………..”

Should be “Adverse events consisted of a mild hepatitis (values up to 3 times

above the normal), a short duration watery diarrhea, a cutaneous maculopapular skin rash and an isolated fever in 1 case each. No renal failure was diagnosed during follow-up.

 SXT was interrupted in 6% (n=3): 1 patient had replacement with a switch to another antimicrobial therapy (n=1) and 2 antibiotic discontinuations were noted. “

Corrected

Discussion

 Better wording would be as follows in the sentences containing the following phrases

 Should be “ This retrospective study documents numerous patients treated with a

combination therapy based on SXT for the treatment of a polymicrobial BJI”

 Should be “ ………. can appear disappointing but it rises up to 87% in the best case scenario.”

 Should be “ This rate is gratifying if we consider the higher proportion than usual of these particular prosthetic joint infections due to polymicrobial and Gram negative bacteria.”

 Should be “ Moreover it concerns elderly and fragile patients with frequent comorbidities who underwent previous surgeries which failed initial therapy and therefore required the expertise of a reference center.”

 Should be “ This latter study showed similar outcomes using this regimen based on SXT versus………..”

 Should be “ In addition, median duration of treatment was 6 weeks which is generally compliant according to the French guidelines [22] and promoting the appropriate use of antibiotic duration.”

Corrected

 Should be “ Likewise, although costs have not been studied, some literature has already reported a diminution of costs when…….” And there should be a reference added to support .

References were added, sorry for that. It was a problem with the software (Mendeley).

The phrase “………but also than the ones reported in previous cohorts of BJI treated with SXT………… “ is missing something – please correct

We rephrased it as follows: “but also than the ones reported in previous cohorts of BJIs treated with SXT which preconized high doses [27,29–31].” 

Should be “ ………. For antibiotic selective pressure,”

Eliminate the sentence “A particular attention to renal function and potassium level must be paid in these patients especially when using high dosing of SXT. “ It adds very little and helps to tighten the manuscript.

Done

Conclusion

 Eliminate “……….in common daily encountered practice………….”

Withdrawn.

References

 There are a few items to correct with respect to case in the references eg the last 2 and other refs have the title of the paper all in capital letters – please correct

Corrected.

---

## [Editor Report · Decision Letter 2]

3 Oct 2019

PONE-D-19-17215R2

Efficacy of cotrimoxazole (Sulfamethoxazole-Trimethoprim) as a salvage therapy for the treatment of bone and joint infections (BJIs)

PLOS ONE

Dear Dr. Davido,

Thank you for submitting your re-revised manuscript to PLOS ONE and carefully making the suggested edits and changes. . Unfortunately there remain a few areas where the text remains unclear and additional minor edits are  be required to ensure that the readership is able to completely follow the points that are being made. Therefore, we invite you to submit a further revised version of the manuscript that addresses the points raised during the most recent editorial review process.

Abstract 

Should be "..... a broad spectrum of activity with adequate...."

"We noted 76% of orthopedic device infection (ODI)."  is a grammatically poor sentence and suggest to add the finding into the first sentence to avoid 2 short choppy sentences. Also in Table 1 you indicate that the age 60 was a median but you indicate a mean in the abstract - please correct  and also add in brackets the SD in the abstract ie " We analyzed 51 patients with a ?  mean ? median  age of 60 +> 20  (SD) years of which 76% presented with an orthopedic device infection (ODI). 

Introduction

Should be "..... due to a broad spectrum of activity with good bone diffusion...."

Italics for *Staphylococcus spp* was forgotten to be added

The genus for *Clostridium* has been officially changed to *Clostridioides *so please change throughout the text where used but not the references

Should be " leading to the suggestion of SXT.... "

Methods

Should be ".....SXT use, route of administration..... " rather than currently shown as "SXT use: route of administration

The sentence ending with "associated drugs , "  ends with a  comma rather than a period.

Results

Note the discrepancy of use of a median age as in Table 1 as opposed to mean in the abstract - please correct 

 Discussion

Why do you say " .......... previous surgeries which failed failure  initial therapy ....." - please clarify or correct

Please correct the sentence " ...Harbarth et al described 85.7% of success ...." do you mean " a 85.7 % success rate...."

Prefer to say " "A combination of therapy of SXT with a  FQ in these patients was associated with a favorable outcome in 82.3% ." than as currently written.

".... preserve FQs from acquired resistance." should be "...... FQs acquiring resistance"

Something is incomplete or erroneous in the sentence ".... but also than the ones reported in previous  cohorts of BJIs treated with SXT which preconized...." - please correct

Should be plural ie "side effects "

Should be " SXT is responsible for various ..." rather than "... of various "

Should be " .... reinforces the usefulness of SXT"

Please rephrase " deadend situations " as not a scientific term

Should be plural ie    ".... physicians should .... "

Conclusion

Rather than ".... taking into consideration such molecule" why not just say " taking into consideration the use of SXT" since that is what your article suggests

We would appreciate receiving your revised manuscript by Oct 21 2019.  To enhance the reproducibility of your results, we recommend that if applicable you deposit your laboratory protocols in protocols.io, where a protocol can be assigned its own identifier (DOI) such that it can be cited independently in the future. For instructions see: http://journals.plos.org/plosone/s/submission-guidelines#loc-laboratory-protocols

We look forward to receiving your revised manuscript.

Kind regards,

John Conly, MD

Academic Editor

PLOS ONE

---

## [Author Response · Author response to Decision Letter 2]

4 Oct 2019

All the typo have now been corrected. 

Thank you for this careful proofreading.

Abstract

Should be "..... a broad spectrum of activity with adequate...."

Already written as suggested.

"We noted 76% of orthopedic device infection (ODI)." is a grammatically poor sentence and suggest to add the finding into the first sentence to avoid 2 short choppy sentences. Also in Table 1 you indicate that the age 60 was a median but you indicate a mean in the abstract - please correct and also add in brackets the SD in the abstract ie " We analyzed 51 patients with a ? mean ? median age of 60 + 20 (SD) years of which 76% presented with an orthopedic device infection (ODI).

Thank you for the example. Actually median was=median so it is another way to show our results. Usually mean are commonly used in abstract, therefore we kept this presentation of our findings.

Introduction

Should be "..... due to a broad spectrum of activity with good bone diffusion...."

Corrected, thank you.

Italics for Staphylococcus spp was forgotten to be added

It’s now in italics, sorry for that mistake.

The genus for Clostridium has been officially changed to Clostridioides so please change throughout the text where used but not the references

Corrected.

Should be " leading to the suggestion of SXT.... "

Corrected.

Methods

Should be ".....SXT use, route of administration..... " rather than currently shown as "SXT use: route of administration

“:” has been changed to “,”

The sentence ending with "associated drugs , " ends with a comma rather than a period.

It was changed to a period as requested.

Results

Note the discrepancy of use of a median age as in Table 1 as opposed to mean in the abstract - please correct 

Median is used in the Table 1 to illustrate IQR as we are dealing with a small sample size study. As clarified above, Median was equal to mean in our work.

 Discussion

Why do you say " .......... previous surgeries which failed failure initial therapy ....." - please clarify or correct

Sorry for the duplicate and repeated word namely “failed failure”. Sentence was also clarified because it was referring to table 1: “previous surgeries which failed initial therapy (as illustrated in Table 1) and therefore required the expertise of a reference center.”

Please correct the sentence " ...Harbarth et al described 85.7% of success ...." do you mean " a 85.7 % success rate...."

Yes, sorry for that misunderstood. It is now corrected.

Prefer to say " "A combination of therapy of SXT with a FQ in these patients was associated with a favorable outcome in 82.3% ." than as currently written.

Thank you for the correction, which has been applied.

".... preserve FQs from acquired resistance." should be "...... FQs acquiring resistance"

Thank you for the correction, which has been applied.

Something is incomplete or erroneous in the sentence ".... but also than the ones reported in previous cohorts of BJIs treated with SXT which preconized...." - please correct

Sorry the word lower was missing in our sentence. We also shortened the sentence because the end was obvious, as follows: “the doses of SXT we used were frequently lower than recommended by the French (3200/640mg) and US guidelines (8 mg/kg of trimethoprim), but also lower than the ones reported in previous cohorts of BJIs treated with SXT”

Should be plural ie "side effects "

 Corrected.

Should be " SXT is responsible for various ..." rather than "... of various "

Thank you for the correction, which has been applied.

Should be " .... reinforces the usefulness of SXT"

Sorry for the typo, it has been corrected.

Please rephrase " deadend situations " as not a scientific term

It has been changed to “deadlock” which can be used in scientific term, especially for antibiotics. (ie : https://www.ncbi.nlm.nih.gov/pubmed/24698433)

Should be plural ie ".... physicians should .... "

 Corrected.

Conclusion

Rather than ".... taking into consideration such molecule" why not just say " taking into consideration the use of SXT" since that is what your article suggests

 Thank you for the suggestion which clarified the conclusion and has been applied.

---

## [Editor Report · Decision Letter 3]

7 Oct 2019

Efficacy of cotrimoxazole (Sulfamethoxazole-Trimethoprim) as a salvage therapy for the treatment of bone and joint infections (BJIs)

PONE-D-19-17215R3

Dear Dr. Davido,

We are pleased to inform you that your manuscript has been judged scientifically suitable for publication and will be formally accepted for publication once it complies with all outstanding technical requirements.

With kind regards,

John Conly, MD

Academic Editor

PLOS ONE
---

## [Editor Report · Acceptance letter]

9 Oct 2019

PONE-D-19-17215R3 

Efficacy of cotrimoxazole (Sulfamethoxazole-Trimethoprim) as a salvage therapy for the treatment of bone and joint infections (BJIs) 

Dear Dr. Davido:

I am pleased to inform you that your manuscript has been deemed suitable for publication in PLOS ONE. Congratulations! Your manuscript is now with our production department. 

With kind regards,

on behalf of

Dr John Conly 

Academic Editor

PLOS ONE